# Targeting Oxidative Stress: Novel Coumarin-Based Inverse Agonists of GPR55

**DOI:** 10.3390/ijms222111665

**Published:** 2021-10-28

**Authors:** Matthias Apweiler, Soraya Wilke Saliba, Jana Streyczek, Thomas Hurrle, Simone Gräßle, Stefan Bräse, Bernd L. Fiebich

**Affiliations:** 1Neuroimmunology and Neurochemistry Research Group, Department of Psychiatry and Psychotherapy, Medical Center-University of Freiburg, Faculty of Medicine, University of Freiburg, D-79104 Freiburg, Germany; matthias.apweiler@uniklinik-freiburg.de (M.A.); wilkesaliba@yahoo.com.br (S.W.S.); jana.streyczek@uniklinik-freiburg.de (J.S.); 2Institute of Organic Chemistry, Karlsruhe Institute of Technology (KIT), D-76131 Karlsruhe, Germany; hurrle.thomas@gmail.com (T.H.); stefan.braese@kit.edu (S.B.); 3Institute of Biological and Chemical Systems-Functional Molecular Systems (IBCS-FMS), Karlsruhe Institute of Technology (KIT), Hermann-von-Helmholtz-Platz 1, D-76344 Eggenstein-Leopoldshafen, Germany; simone.graessle@kit.edu

**Keywords:** oxidative stress, GPR55, coumarin-based compounds, ROS, inverse agonism, 8-Iso-PGF_2α_

## Abstract

Oxidative stress is associated with different neurological and psychiatric diseases. Therefore, development of new pharmaceuticals targeting oxidative dysregulation might be a promising approach to treat these diseases. The G-protein coupled receptor 55 (GPR55) is broadly expressed in central nervous tissues and cells and is involved in the regulation of inflammatory and oxidative cell homeostasis. We have recently shown that coumarin-based compounds enfold inverse agonistic activities at GPR55 resulting in the inhibition of prostaglandin E_2_. However, the antioxidative effects mediated by GPR55 were not evaluated yet. Therefore, we investigated the antioxidative effects of two novel synthesized coumarin-based compounds, KIT C and KIT H, in primary mouse microglial and human neuronal SK-N-SK cells. KIT C and KIT H show antioxidative properties in SK-N-SH cells as well as in primary microglia. In GPR55-knockout SK-N-SH cells, the antioxidative effects are abolished, suggesting a GPR55-dependent antioxidative mechanism. Since inverse agonistic GPR55 activation in the brain seems to be associated with decreased oxidative stress, KIT C and KIT H possibly act as inverse agonists of GPR55 eliciting promising therapeutic options for oxidative stress related diseases.

## 1. Introduction

Oxidative stress has recently gained more attention in the etiology and pathogenesis of neurological and psychiatric diseases such as Alzheimer’s disease (AD), Parkinson’s disease (PD) and depression [1,2]. Together with neuroinflammation, oxidative stress leads to progressive neurodegeneration, the common path of the mentioned diseases [3]. However, treatment of neurological and psychiatric diseases still focuses on symptomatic relief without targeting the underlying causal pathological molecular mechanisms. With an aging demography and rising incidences of AD [4] and PD [5], adequate and causal treatment with mild side-effects gains even more importance. Therefore, evaluating the role of different receptors and pathways in the homeostasis of oxidative stress in the central nervous system (CNS) might offer new opportunities in the therapy of neurological and psychiatric diseases. 

The G-protein coupled receptor 55 (GPR55) was first described in 1999 [6] and is broadly expressed in different areas of the CNS, such as the frontal cortex or the hippocampus [7]. The discovery of the bioactive lipid lysophosphtatidylinositol (LPI) as endogenous GPR55 agonist led to the receptor’s deorphanization [8]. However, besides LPI, several commercially available as well as endogenous ligands show agonistic or antagonistic activity at the GPR55 [7,9]. Endocannabinoids, 2-arachidonoylglycerol, and delta-9-tetrahydrocannabinol (Δ9-THC) for instance, show strong affinities and activation of GPR55, heating up the discussion about GPR55 as potential third cannabinoid-receptor (CB) [7]. Commercially available GPR55 agonists, such as O-1602, and GPR55-antagonists like ML-193 are commonly used in GPR55 research, to evaluate GPR55-specific molecular pathways and effects. 

Besides these widely used GPR55 ligands, coumarin-derivates show antagonistic coupled to inverse agonistic activities on GPR55-dependent neuroinflammatory processes as reported recently [10,11]. Agonistic activities at G-protein coupled receptors (GPCRs) affect downstream signal cascades activated by the receptor, such as the phosphorylation of protein kinases or the translocation of transcriptional factors in the cell’s nucleus. This “On”-state of a receptor can be changed to “Off” spontaneously or by an antagonist, inhibiting competitively or sterically further agonistic binding or spontaneous receptor activation. This “Off”-state of the receptor after binding of an antagonist is equivalent to the receptor’s physiological resting state [12,13]. Antagonists with inverse agonistic activities, as shown for coumarin derivates, are not just setting a receptor to its resting state, they also induce contrary intracellular signaling as provoked by the receptor’s agonists [12]. Inverse agonists selectively activate different pathways of the bound receptors, leading to opposite effects in comparison to classical receptor agonists [14,15]. Therefore, inverse agonism reveals completely new therapeutic options avoiding and reversing pathological receptor activity [12]. On a molecular level, the inversed agonism is explained by differences in the discriminatory activation of G_α_ or G_βγ_ [16] as well as differentiated phosphorylation of GPCRs [13]. KIT 17, a Coumarin-based compound and GPR55 ligand, showed anti-neuroinflammatory effects in LPS-stimulated primary microglia cells probably relying on an inverse agonistic activity at GPR55 for instance [10].

The role of GPR55 in the regulation of cellular oxidative homeostasis is still poorly understood. Since GPR55 regulates protein kinase C (PKC), phosphatidyl-inositol-3-kinase (PI3K), and protein kinase B (Akt) in its downstream signaling, GPR55 activation might enforce nuclear factor erythroid 2-related factor 2 (Nrf2) expression [17,18]. As transcriptional factor, Nrf2 increases the synthesis of antioxidative enzymes and molecules when cells are facing oxidative stress [18]. However, the activation of GPR55 is connected with increased concentrations of Reactive Oxygen Species (ROS) and apoptosis [19]. In human aortic endothelial cells, the GPR55 inverse agonist CID16020046 reversed pro-oxidative effects induced by oxidized low-density lipoprotein (LDL) underlining the potential of inverse agonists of GPR55 to interfere with regulatory processes of oxidative stress [20]. 

There are multiple assays to evaluate concentrations of ROS and oxidative stress, which are commonly used in research. In this current study, we measured 8-Iso-PGF_2α_, a good known and sensitive marker for oxidative stress, to screen for anti-oxidative effects of the coumarin-derivates [1]. High levels of ROS lead to lipid peroxidation in the cells, and oxidized lipids are increasing levels of isoprostanes [21]. Besides 8-Iso-PGF_2α_-levels, we examined the antioxidative capacity with the oxygen radical antioxidant capacity (ORAC) activity assay, which compares antioxidative properties of compounds to a vitamin E analogue named trolox. The well-established assay measures fluorescent degradation after inducing oxygen radical generation and effects of the tested compounds on prevention of the fluorescent degradation [22]. Furthermore, we introduce a modified cell viability assay to evaluate prevention of H_2_O_2_-induced cell death after pretreatment with compounds. 

The role of GPR55 in the pathogenesis of different diseases, such as obesity and diabetes, bone disorders, like osteoporosis, and various cancers has been shown before [9], but the role and possibly therapeutic potential of the GPR55 in CNS disorders such as depression, AD, and PD is still poorly understood [23,24]. Since the involvement of oxidative stress in the pathogenesis of these diseases is part of actual research, antioxidative GPR55-dependent pathways might be a promising target in future pharmacological therapies. 

The current study investigates the antioxidative effects of two novel synthesized coumarin-based compounds named KIT C and KIT H in human neuroblastoma cells as well as in primary mouse microglia. We hypothesize, that KIT C and KIT H ameliorate interleukin (IL)-1β-, lipopolysaccharide (LPS)- and H_2_O_2_-induced oxidative stress in the cells by acting as inverse agonists in the GPR55. 

## 2. Results

### 2.1. Cytotoxic Effects of KIT C and KIT H

To exclude any cytotoxic effects of the coumarin-based compounds KIT C and KIT H, we performed a MTT-assay to determine the non-toxic concentrations of the compounds. In IL-1β-stimulated SK-N-SH cells, neither KIT C nor KIT H (1–10 µM) significantly decreased the cell viability compared to untreated cells (Figure 1). The solvent of KIT C and KIT H, DMSO, significantly reduced cell metabolism (*p* < 0.05), whereas 5 and 10 µM of KIT C (*p* < 0.0001) and 1 µM of KIT H (*p* < 0.05) significantly increased cell metabolism in the MTT-assay. Since KIT C and KIT H did not reveal cell toxic effect, the tested concentrations up to 10 µM were used for further experiments. 

### 2.2. Effects of KIT C and KIT H on 8-Iso-PGF_2α_-Release in SK-N-SH and Primary Microglial Cells

Next, the effects of KIT C and KIT H on the release of 8-Iso-PGF_2α_ in IL-1β-stimulated SK-N-SH cells (Figure 2A) and LPS-stimulated primary mouse microglia cells (Figure 2B) were evaluated using a commercial 8-Iso-PGF_2α_ enzyme immunoassay (EIA). Release of 8-Iso-PGF_2α_ is strongly induced by the stimulation with IL-1β or LPS. In primary microglia cells (Figure 2B), the basal release of 8-Iso-PGF_2α_ was lower compared to SK-N-SH cells (Figure 2A). Ten µM of KIT C significantly reduced 8-Iso-PGF_2α_-release about 60% in SK-N-SH (*p* < 0.0001) as well as in microglia cells (90% reduction, *p* < 0.0001) to levels comparable to the untreated control. The 30% reduction of 8-Iso-PGF_2α_ levels after pre-treatment of SK-N-SH cells with 10 µM KIT H (*p* < 0.05) was lower compared to KIT C, but still significant compared to IL-1β-stimulated samples. The effects of KIT C and KIT H on 8-Iso-PGF_2α_ levels suggest antioxidative properties of both coumarin-based compounds. Therefore, further experiments were conducted to prove the antioxidative effects and evaluate possible underlying mechanisms. 

### 2.3. Antioxidative Capacity of KIT C and H in the ORAC Assay

The ORAC-assay determines the antioxidant capacity of compounds ex vitro. The raw values of the tested compounds are compared to a trolox standard curve (Figure 3) and can be calculated as Trolox Equivalents (TE; Table 1). Both, KIT C and KIT H, showed antioxidative capacities in the ORAC assay. The maximal antioxidant capacity was achieved using 10 µM KIT C being comparable to 6.5 µM of trolox. Ten µM of KIT H showed antioxidant capacity comparable to 4.7 µM of Trolox. Interestingly, the antioxidant capacity of KIT C and KIT H is not correlated with the trolox standard curve but seems to reach its plateau at concentrations around 10 µM. The antioxidant capacity of KIT C and KIT H is, therefore, the maximal half of the trolox antioxidative capacity when comparing the concentrations of 10 µM of KIT C and KIT H. 

### 2.4. GPR55-Dependent Antioxidative Effects

A modified MTT-assay was designed and conducted to evaluate concentration-dependent effects of KIT C and KIT H on ROS-induced cell death in SK-N-SH cells (Figure 4). Figure 4 demonstrates prevention of cell death induced by oxidative stress with negative spikes compared to the baseline. Positive values represent increased cell death. Two hundred fifty µM of H_2_O_2_ showed less induction of cell death compared to 500 µM H_2_O_2_. Twenty µL of ethanol (approximately 20% EtOH final concentration) reliably induced cell death (*p* < 0.0001). KIT C concentration-dependent and significantly decreased ROS-induced cell death about maximal 30% with maximal effects using 5 and 10 µM (*p* < 0.0001 and *p* < 0.01). KIT H showed less efficient concentration-dependent prevention of ROS-induced cell death compared to KIT C, with a significant reduction at concentrations of 10 µM of KIT H and maximal inhibition of around 23% (*p* < 0.05). 

### 2.5. GPR55-Dependent Antioxidative Effects in GPR55 Knock Out Cells

The effects of KIT C and KIT H on ROS-induced cell death were re-evaluated in GPR55-knockout SK-N-SH cells (Figure 5). Again, H_2_O_2_ concentration-dependently induced cell death and ethanol (*p* < 0.0001) strongly reduced cell viability as well. The effects of KIT C and KIT H on ROS-induced cell death were abolished in the GPR55-knockout-SK-N-SH cells, with no significant inhibitory effects of both coumarin-based compounds. 

### 2.6. Nrf-2 Expression in SK-N-SH Cells

Next, we investigated potential underlying mechanisms of the antioxidative effects of KIT C and KIT H using qPCR. IL-1β treatment for 4 h induced Nrf2 expression in SK-N-SH cells as compared to untreated cells (Figure 6). KIT C in the concentrations 0.1 (*p* < 0.01) and 10 µM (*p* < 0.0001) and KIT H in the highest concentration of 10 µM (*p* < 0.01) significantly enhanced IL-1β-induced Nrf2 expression. Furthermore, we evaluated effects of KIT C and KIT H on the expression of superoxide dismutase (SOD)-1 and SOD-2 as two key enzymes in antioxidative cell mechanisms (data not shown). Neither SOD-1 nor SOD-2 expressions were affected by KIT C and KIT H.

## 3. Discussion

In this study, we investigated the antioxidative capacities of two novel coumarin-based compounds, KIT C and KIT H. Both reduced 8-Iso-PGF_2α_ and thus lipid peroxidation in IL-1β-treated SK-N-SH cells, and KIT C but not KIT H in LPS-stimulated primary mouse microglia. Both coumarin-based compounds prevented H_2_O_2_-induced cell death in SK-N-SH cells with antioxidative capacities maximal half of Trolox’s antioxidative capacity. The prevention of ROS-induced cell death was abolished by knockout of GPR55 in SK-N-SH cells. Nrf2 expression was upregulated after treatment with KIT C and KIT H in concentration of 10 µM in IL-1β-stimulated SK-N-SH cells, but SOD-1/2 expression was not significantly affected by KIT C or KIT H. 

8-Iso-PGF_2α_ is known as reliable marker for oxidative cellular stress [25], therefore KIT C- and KIT H-dependent decrease in 8-Iso-PGF_2α_ in SK-N-SH cells strongly suggests antioxidative properties of both compounds. KIT H shows less antioxidative effects than KIT C in SK-N-SH cells and in LPS-stimulated mouse primary microglia. KIT H, in contrast to KIT C, did not significantly reduce 8-Iso-PGF_2α_-formation in primary microglia. The differences of the activities of KIT C and KIT H in human SK-N-SH cells and primary microglia might be explained by the fact that GPR55 in mice and human exhibits minor differences in the binding pocket, possibly leading to the differences in the antioxidative capacities of KIT H in SK cells and primary murine microglia. The aromatic ring of KIT H instead of the methyl-group of KIT C might explain the differences between KIT H and KIT C in respect to their antioxidative effects on lipid peroxidation in both cell types. This is supported by data showing that classical agonists of GPR55 lack the selective antagonist-typical phenolic or heterocyclic group reaching out of the GPR55 binding pocket. This additional cyclic group of GPR55 antagonists prevents conformational change of the receptor and in consequence, the activation of coupled signal pathways [9]. Most GPR55 antagonists are not reaching as deep in the receptor’s binding pocket and in consequence do not activate the receptor and its signaling and therefore only reveal antagonistic activities. Compounds with a remaining thin vertical structure in the binding pocket, as suggested for KIT C and KIT H, and without associated cyclic functional groups therefore can activate receptors by binding deeper in the binding pocket [9] being still antagonists with inverse agonistic activity. Since KIT H has an additional phenolic ring, it might show more antagonistic properties and less inverse agonistic activity than KIT C. 

Nevertheless, KIT H still prevents ROS-induced stress in a nearly comparable potency to KIT C in the ROS-MTT-assay. On the one hand, its effects on lipid peroxidation might differ from its effects on the clearance of reactive oxygen species (ROS) due to different activation of antioxidative enzymes by the compounds. On the other hand, KIT H might show antiapoptotic effects, interacting with the introduced ROS-MTT assay. This leads to an interpretation problem of the ROS-MTT assay, which allows only to draw indirect conclusions about antioxidative mechanisms involved. Since the MTT-assay itself measures cell viability indirectly by determining cell metabolism, the observed effects can be affected by changes in cellular metabolism, apoptotic, or antioxidative pathways. For these reasons, the ROS-MTT assay should be interpreted only together with other oxidative assays, such as the 8-Iso-PGF_2α_-EIA or the ORAC-assay as presented here. The ORAC-assay explains the comparable effects of KIT C and KIT H on H_2_O_2_-induced cell death in the ROS-MTT assay as well. The concentration of 10 µM shows similar antioxidative capacities of KIT C and KIT H in comparison to Trolox, so one would expect comparable antioxidative effects in the assays used. Another assay often published to investigate oxidative stress is the DCFH-DA-assay. However, in SK-N-SH cells, the assay did not show consistent and comparable results between different measurements, so we were not able to draw reliable and conclusive statements using the DCFH-DA assay. 

It has been shown before, that coumarin-based compounds act as ligands on the GPR55 [10,11,26]. However, the receptor binding profile of the novel coumarin-based compounds KIT C and KIT H was not evaluated yet, for example by radioligand assays. To confirm that the observed antioxidative effects by KIT C and KIT H are mediated via GPR55 receptors, we knocked out the GPR55 in SK-N-SH cells using a commercial CRISPR/Cas9 system and repeated the ROS-MTT assay in the knockout cell line. In line with the previous results of coumarin-based compounds [10,11], KIT C and KIT H exerted its effects by binding to GPR55, since the observed antioxidative effects of both compounds on ROS-induced cell death were abolished in GPR55-knockout cells. 

As outlined before, GPR55 is involved in the regulation of cellular oxidative homeostasis. Activation of GPR55 possibly promotes generation of ROS, since p38 mitogen-activated protein kinase (p38 MAPK) and intracellular calcium effluxes, as described down-stream signaling of this receptor, induce mitochondrial ROS generation [27]. In line with these considerations, the GPR55 agonist *N*-docosahexaenoyl dopamine (DHA-DA) induced apoptosis upon NO- and ROS-production in non-differentiated PC12-cells. The GPR55 antagonist O-1918 reduced this DHA-DA-dependent ROS-production [19]. Furthermore, the inverse agonist of GPR55 CID16020046 showed antioxidative effects in oxidized low density lipoprotein (oxLDL)-induced endothelial inflammation in human aortic cells [20]. The interpretation of the current study’s results supports the view that GPR55 activation is pro-oxidative. Therefore, classical GPR55 agonists seem to have the potential to induce oxidative stress. Since KIT C and KIT H, GPR55-dependently reversed IL-1β-induced lipid peroxidation and H_2_O_2_-induced cell death, both coumarin-based compounds are likely to act as inverse agonists of GPR55 with activities comparable to CID16020046. In contrast, corticosterone-induced inflammatory and oxidative damage in the urinary bladder of Wistar rats, measured by increased concentrations of malondialdehyde and 3-nitrotyrosine, was reduced by treatment with the GPR55 agonist O-1602 [28]. However, GPR55 activation might lead to different effects depending on the examined organisms, tissues, or cells. Further studies are necessary to elucidate the role of GPR55 in pro- and antioxidative processes and the underlying pathways and mechanisms. 

KIT C as well as KIT H increased IL-1β-induced Nrf2-expression in the highest concentration used. Nrf2 is known as central regulator of antioxidative cell reactions, acting as transcriptional factor if cells are confronted with oxidative stress [29]. Therefore, the mechanism of KIT C and KIT H leading to decreased antioxidative stress are possibly Nrf2-dependent. However, investigating the expression of genes regulated by Nrf2, such as SOD-1 and SOD-2, did not explain the antioxidative effects shown in the study. Since Nrf2 is not only regulated on mRNA but also on protein and phosphorylation levels, with inhibitory proteins binding to Nrf2 [30], future studies elucidating the underlying mechanisms of the antioxidative effects of KIT C and KIT H on protein and activation levels of the complex Nrf2 pathway are necessary. 

We show here promising antioxidative properties of KIT C and KIT H in human neuroblastoma cells and primary mouse microglia in vitro. Therefore, coumarin-based compounds might be interesting options in the treatment of diseases associated with oxidative stress, such as depression and other neurological and psychiatric diseases. However, further research needs to be conducted to understand underlying mechanisms of the antioxidative effects and the importance of the effects in animals’ disease models. Ex vivo and in vivo studies are necessary to endorse the shown effects in cell lines and primary microglia in a more complex organism. Besides the demonstrated antioxidative effects, further research on KIT C and KIT H should focus on neuroinflammatory effects of the novel coumarin-based compounds. For other coumarin-scaffolds, GPR55-dependent neuroinflammatory effects have already been shown [10,11]. 

## 4. Materials and Methods

### 4.1. Chemicals

KIT C and KIT H (Figure 7) were synthesized at the Institute of Organic Chemistry, Karlsruhe Institute of Technology (KIT) and provided for further experiments. 

**Synthesis of KIT C (8-Isopropyl-3,5-dimethyl-2*H***-**chromen-2-one):** 2-Hydroxy-6-methyl-3-propan-2-ylbenzaldehyde (150 mg, 840 µmol, 1.00 equiv.), 380 µL propionic acid anhydride (383 mg, 2.95 mmol, 3.50 equiv.) and 5.82 mg of potassium carbonate (40.0 µmol, 0.05 equiv.) were placed in a microwave vial and heated at 180 °C for 65 min at 300 W microwave irradiation. The resulted mixture was allowed to cool to room temperature, poured onto crushed ice and the pH was adjusted to ~7 with sodium bicarbonate. The mixture was then extracted with diethyl ether and the organic phase dried over sodium sulfate and the volatiles were removed under reduced pressure to result in 182 mg (quant.) of an off-white solid. The full description of the reaction was deposited to the repository Chemotion and can be retrieved under the accession number CRR-9502 [31].

*R*_f_ (CH/EtOAc 20:1): 0.33. MP: 148.5 °C. ^1^H NMR (400 MHz, CDCl_3_, ppm) δ = 7.71 (q, *J* = 1.3 Hz, 1H, 4-CH), 7.27 (d, *J* = 7.7 Hz, 1H, ArH), 7.03 (d, *J* = 7.8 Hz, 1H, ArH), 3.59 (hept., *J* = 6.9 Hz, 1H, CH(CH_3_)_2_), 2.47 (s, 3H, ArCH_3_), 2.24 (d, *J* = 1.3 Hz, 3H, C-3(CH_3_), 1.27 (d, *J* = 11.2 Hz, 6H, CH(CH_3_)_2_). ^13^C NMR (100 MHz, CDCl_3_, ppm) δ = 162.4 (C_q_, C-2), 151.1 (C_q_, C_Ar_O), 136.9 (+, CH, C-4), 134.2 (C_q_, C_Ar_), 132.3 (C_q_, C_Ar_), 127.3 (+, C_Ar_H), 125.5 (+, C_Ar_H), 124.6 (C_q_, C_Ar_), 118.2 (C_q_, C-3), 26.4 (+, CH(CH_3_)_2_), 22.9 (+, 2 × CH_3_, CH(CH_3_)_2_), 18.3 (+, CH_3_), 17.5 (+, CH_3_). − IR (KBr): ῦ = 2958 (w), 2925 (vw), 1709 (w), 1594 (w), 1489 (vw), 1448 (w), 1377 (w), 1360 (w), 1293 (vw), 1259 (w), 1186 (w), 1087 (w), 1062 (w), 1046 (w), 1003 (w), 897 (w), 835 (w), 771 (w), 746 (w), 719 (vw), 633 (vw), 607 (w), 480 (w), 410 (vw) cm^−1^. MS (70 eV, EI): *m/z* (%) = 216 (40) [M]^+^, 202 (15), 201 (100) [M − CH_3_]^+^, 173 (5), 130 (7), 129 (11), 128 (13), 115 (11), 91 (6). HRMS (C_14_H_16_O_2_): calc. 216.1145, found 216.1145. Elemental analysis: C_14_H_16_O_2_ (216.18): calc. C 77.75, H 7.46, found C 77.34, H 7.47. The full record of the analyses was deposited to the repository Chemotion and can be retrieved under the accession number CRS-9508 [32]. Reference material was also submitted and can be requested with the same accession number/DOI. 

**Synthesis of KIT H (****3-Benzyl-8-isopropyl-5-methyl-2*H*-chromen-2-one):** Under argon atmosphere, 2-Hydroxy-6-methyl-3-propan-2-ylbenzaldehyde (150 mg, 0.842 mmol, 1.00 equiv.), 140 mg of potassium carbonate (1.01 mmol, 1.20 equiv.), 281 mg of cinnamic aldehyde (2.01 mmol, 2.50 equiv.) and 224 mg of 1,3-dimethylimidazolium dimethyl phosphate (1.01 mmol, 1.20 equiv.) were suspended in 2.50 mL of toluene. The reaction mixture was heated at 110 °C for 50 min via microwave irradiation (maximum pressure of 7 bar). After cooling to room temperature, 5.0 mL of water was added, and the mixture was extracted with 3 × 15 mL of ethyl acetate. Removal of the volatiles under reduced pressure and purification via flash column chromatography (CH/EtOAc 20:1) resulted in a mixture of the product and a methyl ester impurity as a solid. The mixture was stirred at room temperature for 30 min in 80 mL of 1 M sodium hydroxide-solution resulting in a dispersion. The remaining solid was dissolved in ethyl acetate and the organic layer was separated and washed again with 40 mL of 1 M sodium hydroxide-solution and then dried over sodium sulfate. Removal of the volatiles gave 100 mg (41%) of the pure product as a colorless solid. The full description of the reaction was deposited to the repository Chemotion and can be retrieved under the accession number CRR-9509 [33]. 

*R*_f_ (CH/EtOAc 20:1): 0.45. MP: 146.4 °C − ^1^H NMR (400 MHz, CDCl_3_, ppm) δ = 7.54 (t, *J* = 1.2 Hz, 1H, 4-CH), 7.38 − 7.24 (m, 6H, 6 × H_Ar_), 7.02 (d, *J* = 7.9 Hz, 1H, *H*_Ar_), 3.92 (s, 2H, BnC*H*_2_), 3.59 (hept, *J* = 6.9 Hz, 1H, CH(CH_3_)_2_), 2.37 (s, 3H, CH_3_), 1.27 (d, *J* = 6.9 Hz, 6H, CH(CH_3_)_2_). ^13^C NMR (100 MHz, CDCl_3_, ppm) δ = 161.8 (C_q_, C_Ar_), 151.0 (C_q_, C_Ar_), 138.2 (C_q_, C_Ar_), 137.0 (+, C_Ar_H), 134.2 (C_q_, C_Ar_), 132.8 (C_q_, C_Ar_), 129.3 (+, 2 × C_Ar_H), 128.8 (+, 2 × C_Ar_H), 128.1 (C_q_, C_Ar_), 127.7 (+, C_Ar_H), 126.8 (+, C_Ar_H), 125.5 (+, C_Ar_H), 118.0 (C_q_, C_Ar_), 36.9 (−, CH_2_), 26.4 (+, CH), 22.9 (+, 2 × CH_3_, CH(CH_3_)_2_), 18.2 (+, CH_3_). IR (KBr): ῦ = 3026 (vw), 2959 (w), 1699 (m), 1636 (w), 1598 (w), 1489 (w), 1453 (w), 1304 (w), 1258 (w), 1207 (w), 1170 (m), 1068 (m), 1033 (m), 997 (w), 918 (w), 892 (w), 846 (w), 828 (w), 771 (w), 754 (w), 722 (w), 698 (m), 664 (w), 623 (w), 610 (w), 568 (w), 498 (w) cm^−1^. MS (70 eV, EI): *m/z* (%) = 292 (100) [M]^+^, 278 (16), 277 (70) [M − CH_3_]^+^, 171 (13), 128 (6), 115 (8), 91 (48). HRMS (C_20_H_20_O_2_): calc. 292.1458, found 292.1457. Elemental analysis: C_20_H_20_O_2_ (292.15): calc. C 82.16, H 6.90, found C 81.82, H 6.98. The full record of the analyses was deposited to the repository Chemotion and can be retrieved under the accession number CRS-9514 [34]. Reference material was also submitted and can be requested with the same accession number/DOI.

KIT C and KIT H were dissolved in DMSO (Merck KGaA, Darmstadt, Germany) and used in final concentrations of 0.1–10 µM. 

### 4.2. Other Materials

Human interleukin (IL)-1β (100,000 U/mL in phosphate buffered saline (PBS) from Roche Diagnostics, Manheim, Germany) was used at a final concentration of 10 U/mL in SK-N-SH cell cultures. 5 mg/mL lipopolysaccharide (LPS) from Salmonella typhimurium (Sigma Aldrich, Deissenhofen, Germany) was dissolved in PBS as stock and used at a final concentration of 10 ng/mL in primary microglia cultures. 

### 4.3. Human Neuroblastoma (SK-N-SH) Cell Culture

SK-N-SH neuroblastoma cells were obtained from the American Type Culture Collection (HTB-11, Rockville, ML, USA) and grown in 1 × minimum essential medium (MEM) containing Earl’s salts, 10% fetal bovine serum (Bio & SELL GmbH, Feucht/Nürnberg, Germany), 1 mM L-glutamine, 1 mM sodium pyruvate, 4 mL/L of 100 × MEM vitamin solution, 40 units/mL penicillin, 40 µg/mL streptomycin, 0.1 µg/mL fungizone^®^ (all obtained from Gibco, Thermo Fisher Scientific, Bonn, Germany). The cells were incubated at 37 °C with 5% CO_2_ in a humidified atmosphere. Confluent monolayers were passaged routinely by trypsinization. After trypsinization, cells were harvested and re-seeded into 6-, 12-, 24- or 96-well plates. The next day, medium was changed and after 1 h, cells were stimulated for respective experiments.

### 4.4. GPR55-Knockout with CRISPR/Cas9 System in SK-N-SH Cells

For the knockout of the GPR55 in SK-N-SH cells, a modified CRISPR/Cas9 method was used. Commercially available GPR55 Double Nickase Plasmids (sc-411265-NIC, Santa Cruz Biotechnology, Inc., Heidelberg, Germany) were used for single-strand GPR55-knockdown with the offered chemical transfection system. Each plasmid coded either for the green fluorescent protein (GFP) or a puromycin-resistance for selecting successfully transfected cells after transfection. The GPR55-knockout was performed strictly following the manufacturer’s protocols. Briefly, cells were seeded to a 6-well plate and grown to approximately 60% confluency. The combination of 2 µg GPR55 Double Nickase Plasmid and 10 µL UltraCruz^®^ Transfection Reagent showed the best transfection and GPR55-knockdown results after 3 days of incubation in antibiotic-free medium. Afterwards, puromycin-resistant cells were selected by adding puromycin with a final concentration of 1 µg/mL to the wells, which killed all remaining SK-N-SH wildtype cells after three more days of incubation. Then, GFP-expression was checked using a fluorescence microscope and qPCR for GPR55 was performed with the successfully transfected cells to prove GPR55 deficiency. The GPR55-knockdown SK-N-SH cells were grown as described for wildtype SK-N-SH cells and used for further experiments. 

### 4.5. Primary Mouse Microglial Cultures

#### 4.5.1. Ethics Statement

Animals were obtained from the Center for Experimental Models and Transgenic Services-Freiburg (CEMT-FR). All the experiments were approved and conducted according to the guidelines of the ethics committee of the University of Freiburg Medical School under protocol No. X-19/06R and the study was carefully planned to minimize the number of animals used and their suffering. 

#### 4.5.2. Primary Mouse Microglia Cultures

Primary mouse mixed glia cultures were prepared from 2 to 3 days old C57B1/6 WT mice as described before [35]. Briefly, brains were carefully taken under sterile conditions and meninges were removed. The cortices were dissociated and filtered through a 70 µm nylon cell strainer (BD biosciences, Heidelberg, Germany). After centrifugation at 1000 rpm for 10 min the cells were resuspended in Dulbecco’s modified Eagle’s medium (DMEM) with 10% fetal calf serum (FCS; Bio&SELL GmbH, Nürnberg/Feucht, Germany) and antibiotics (DMEM and anti-anti obtained from Gibco, Thermo Fisher Scientific, Bonn, Germany) and cultured in 10 cm cell culture dishes (Falcon, Heidelberg, Germany) in humified atmosphere at 10% CO_2_ and 37 °C. After 12 days in vitro, floating microglia were harvested and re-seeded into 75 cm^2^ culture flasks to obtain pure microglial cultures and plated for further experiments after further growing. The next day, medium was changed to remove non-adherent cells, and after 1 h, cells were stimulated for the experiments. 

### 4.6. Cell Viability Assay

Viability of SK-N-SH cells after treatment with KIT C and KIT H was determined by MTT assay (Sigma-Aldrich GmbH, Taufkirchen, Germany). This assay quantifies the number of metabolically active and viable cells in cell culture based on the reduction of a yellow tetrazolium salt (3-(4,5-dimethylthiazol-2-yl)-2,5-diphenyltetrazolium bromide or MTT) to purple formazan in the cells. Briefly, cells were cultured on 96-well plates at the density of approximately 25 × 10^3^ cells/well for 24 h. Then, medium was changed and after at least 1 h, cells were pre-treated with different concentrations of KIT C (0.1–10 μM) or KIT H (0.1–10 μM) for 30 min. Cells were then incubated with or without IL-1β for the next 20 h. Ethanol (20% end conc.; Sigma-Aldrich GmbH, Taufkirchen, Germany) was used as a positive control to induce cell death. Next, 20 µL of MTT-solution (5 mg/mL) were added to all wells and cells were incubated for another 4 h at 37 °C. After that, medium was removed and replaced with 200 µL of DMSO for cell lysis. Colorimetric reaction was measured using MRX^e^ Microplate reader (Dynex Technologies, Denkerdorf, Germany) at 595 nm.

### 4.7. Determination of 8-Iso-PGF_2α_ (8-Isoprostane) Release

SK-N-SH cells were pre-treated with KIT C or KIT H (0.1–10 µM) for 30 min. Afterwards, cells were incubated with or without IL-1β (10 U/mL) for the next 24 h and supernatants were collected. The levels of 8-Iso-PGF_2α_ (8-isoprostane) were measured using a commercially available enzyme immunoassay (EIA) kit (from Cayman Chemicals, Ann Arbor, MI, USA, distributed by BioMol, Hamburg, Germany) following the manufacturer’s protocol. The results were normalized to IL-1β and presented as percentage of change in PGs levels of at least three independent experiments. Primary microglia cultures were treated like SK-N-SH cells, except for 24 h LPS-, instead of IL-1β-stimulation. Again, supernatants were collected after 24 h and used for the 8-Iso-PGF_2α_-EIA. The results were normalized to LPS and presented as a percentage change in PGs levels of at least three independent experiments.

### 4.8. ROS-MTT Assay

We modified the MTT assay used to investigate cell viability to examine the prevention of oxidative stress-induced cell death after treatment of SK-N-SH cells with H_2_O_2_. The assay was designed based on a publication using the alamarBlue™ assay to evaluate cell viability after treatment with pro- and antioxidants [36]. Prevention of ROS-induced cell death should allow us to draw indirect conclusions about antioxidative properties of the tested compounds. The assay was carried out like the normal MTT assay described before but without IL-1β being used for cell stimulation after pre-treatment of the cells with KIT C or KIT H. Instead, 250 µM or 500 µM H_2_O_2_ were administered to the wells, inducing oxidative stress. To establish the best duration of stimulation for the maximum induced ROS-dependent cell death, we compared 24 and 48 h of H_2_O_2_-incubation. Since we did not observe momentous differences between those stimulation times, we conducted the ROS-MTT assay with 24 h of stimulation. Ethanol was used as a positive control, reliably inducing cell death. 

### 4.9. RNA Isolation and Quantitative PCR

For quantification of the mRNA of the enzymes of the Nrf2 pathway, we performed quantitative real-time PCR (qPCR) in SK-N-SH cells. Cultured cells were pre-treated with KIT C (0.1–10 µM) or KIT H (0.1–10 µM) for 30 min, followed by stimulation with IL-1β (10 U/mL) for 4 h. RNA was extracted using the GeneMATRIX Universal RNA Purification Kit (Roboklon GmbH, Berlin, Deutschland) according to the manufacturer’s protocol. Then, cDNA was reverse transcribed from 500 ng of total RNA in a 30 μL total reaction volume with initial denaturation at 70 °C (10 min) with a following amplification cycle after addition of 10 µL master mix. The qPCR amplification was carried out with the CFX96 real-time PCR detection system (Bio-Rad Laboratories GmbH, Feldkirchen, Germany). Glyceraldehyde 3-phosphate dehydrogenase (GAPDH) served as an internal control for sample normalization. The primer sequences used were GAPDH: Fwd: 5′-TGGGAAGCTGGTCATCAAC-3′/Rev: 5′-GCATCACCCCATTTGATGTT-3’ and Nrf2: Fwd 5′-ACACGGTCCACAGCTCATC-3′/Rev 5′-TGCCTCCAAAGTATGTCAATCA-3′. Primers were designed using Universal ProbeLibrary Assay Design Center (Roche Diagnostics, Mannheim, Germany) and obtained by biomers.net GmbH (Ulm, Germany).

### 4.10. ORAC-Assay

The antioxidative capacities of KIT C and KIT H (1 and 10 µM) were evaluated using the OxiSelect™ oxygen radical antioxidant capacity (ORAC) ex vivo activity assay (Cell Biolabs, Inc., San Diego, CA, USA) following the manufacturer’s instructions. Trolox, an analog of vitamin E but soluble in water, is known for its antioxidative effects in higher concentrations and therefore is often used as positive control in antioxidative assays. Briefly, Trolox™ antioxidant standard or of KIT C and KIT H were added to a 96-well plate, mixed with fluorescein solution and after 30 min incubation at 37 °C free radical initiator solution was added to all wells. Fluorescence was determined at 37 °C using a microplate reader (PerkinElmer Victor X5 2030-0050 Multimode Plate Reader, Rodgau, Germany; excitation wavelength 485 nm, emission wavelength 535 nm). Raw values were transformed to Trolox Equivalents (TE) after calculating the area under the curve (AUC) for each concentration of KIT C or KIT H. 

### 4.11. Statistical Analysis

Raw values were converted to percentage and IL-1β (10 U/mL), LPS (10 ng/mL), or the appropriate positive control, such as untreated cells for MTT-assay, were considered as 100%. Data of the ROS-MTT assay are shown as changes in cell viability considering 500 µM H_2_O_2_ (+1 µL DMSO) as 0%. Data are represented as mean ± SD of at least three independent experiments. Statistical comparisons were performed using one-way ANOVA with Dunett’s post hoc test (Prism 8 software, GraphPad software Inc., San Diego, CA, USA). The level of significance was set at * *p* < 0.05, ** *p* < 0.01, *** *p* < 0.001 and **** *p* < 0.0001.

## 5. Conclusions

KIT C and KIT H exert strong antioxidative effects in SK-N-SH cells, with KIT C having a greater impact on lipid peroxidation than KIT H. The ORAC-assay proves that the antioxidative capacities of both compounds are also represented in the ROS-MTT assay. Both compounds prevent H_2_O_2_-induced cell death, being abolished after knocking out GPR55 in SK-N-SH cells. Therefore, targeting GPR55 with inverse agonists, such as KIT C and KIT H, enfolds promising antioxidative effects. Since growing evidence suggests association of oxidative stress with different neurodegenerative diseases, further research on these compounds and GPR55 might be beneficial in future therapy of those diseases. 

## Figures and Tables

**Figure 1 ijms-22-11665-f001:**
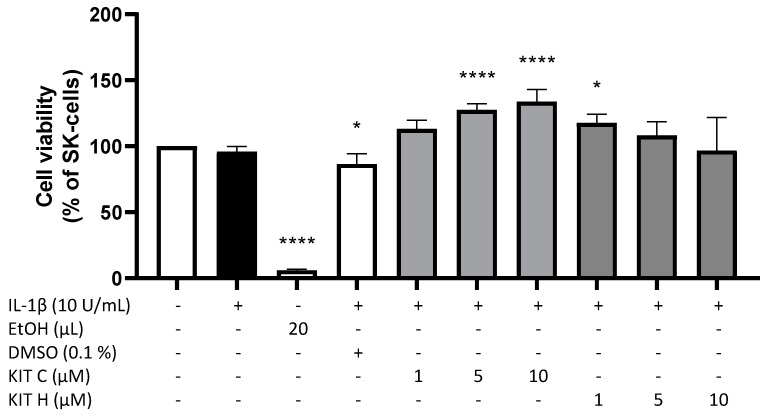
Effects of KIT C (light grey) and KIT H (dark grey) on cell viability of IL-1β-stimulated SK-N-SH cells. Cell viability was measured after 24 h of treatment by change in color due to MTT-oxidation. Values are presented as the mean ± SD of four independent experiments. Statistical analysis was performed using one-way ANOVA with Dunnett’s post hoc test with * *p* < 0.05 and **** *p* < 0.0001 compared to untreated cells.

**Figure 2 ijms-22-11665-f002:**
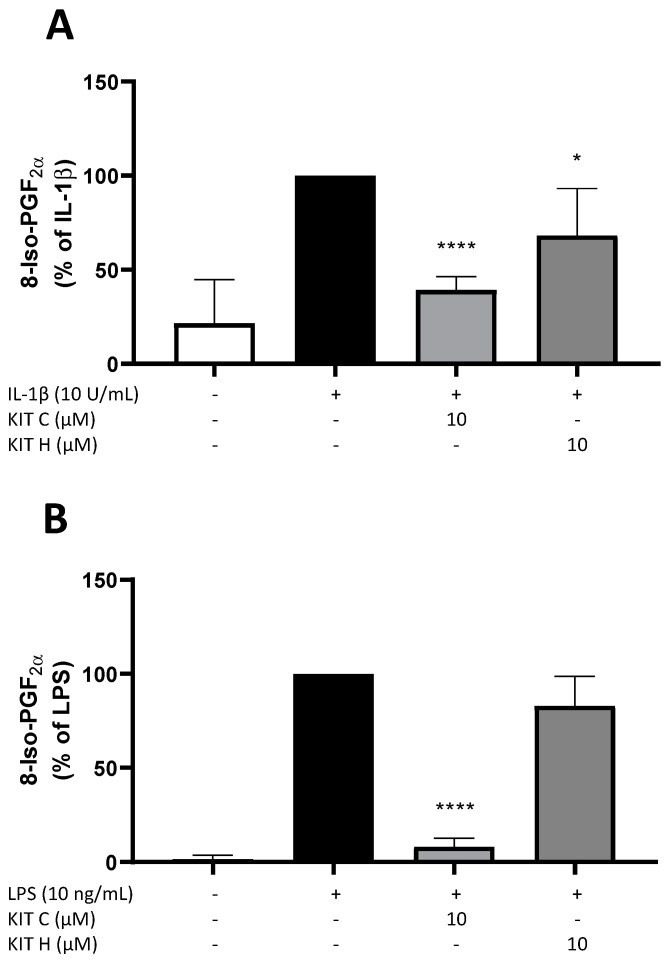
Effects of KIT C (light grey) and KIT H (dark grey) on the release of 8-Iso-PGF_2α_ in IL-1β-stimulated SK-N-SH cells (**A**) and LPS-induced primary mouse microglial cells (**B**). Cells were stimulated as described in material and methods. After 24 h, supernatants were collected and release of 8-Iso-PGF_2α_ was measured by EIA. Values are presented as the mean ± SD of at least three independent experiments. Statistical analysis was performed using one-way ANOVA with Dunnett’s post hoc tests with * *p* < 0.05 and **** *p* < 0.0001 compared to IL-1β or LPS.

**Figure 3 ijms-22-11665-f003:**
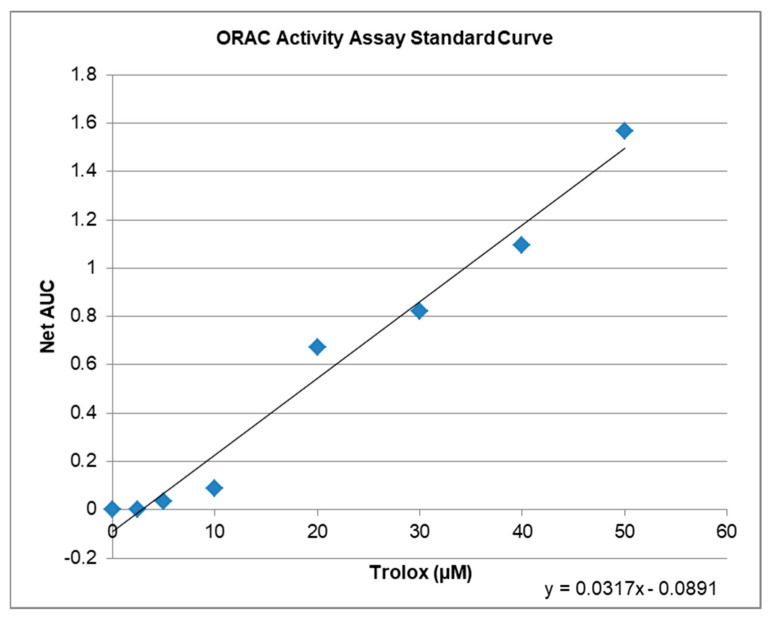
ORAC activity assay standard curve with regression equation y = 0.0317x − 0.0891, that was used to calculate the µM Trolox™ Equivalents (TE) of the tested compounds.

**Figure 4 ijms-22-11665-f004:**
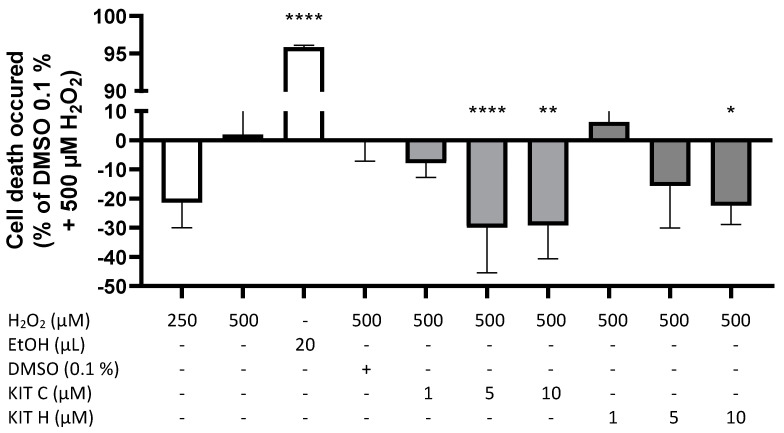
Effects of KIT C (light grey) and KIT H (dark grey) on H_2_O_2_-induced cell death in the ROS-MTT assay in SK-N-SH cells. SK-cells were stimulated as described in material and methods. After 24 h, cell death was measured by change in color due to MTT-oxidation. Values are presented as the mean ± SD of at least three independent experiments. Statistical analysis was performed using one-way ANOVA with Dunnett’s post hoc tests with * *p* < 0.05, ** *p* < 0.01 and **** *p* < 0.0001 compared to 500 µM H_2_O_2_ with 0.1% DMSO.

**Figure 5 ijms-22-11665-f005:**
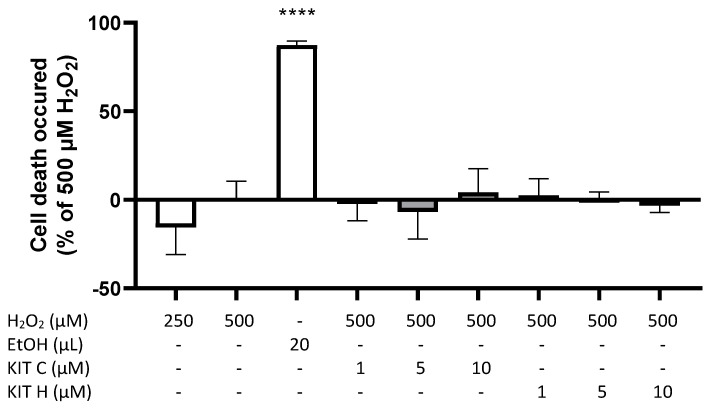
Effects of KIT C (light grey) and KIT H (dark grey) on H_2_O_2_-induced cell death in the ROS-MTT assay in GPR55-knockout SK-N-SH cells. SK-cells were stimulated as described in material and methods. After 24 h, cell death was measured by a change in color due to MTT-oxidation. Values are presented as the mean ± SD of at least three independent experiments. Statistical analysis was performed using one-way ANOVA with Dunnett’s post hoc tests with **** *p* < 0.0001 compared to 500 µM H_2_O_2_.

**Figure 6 ijms-22-11665-f006:**
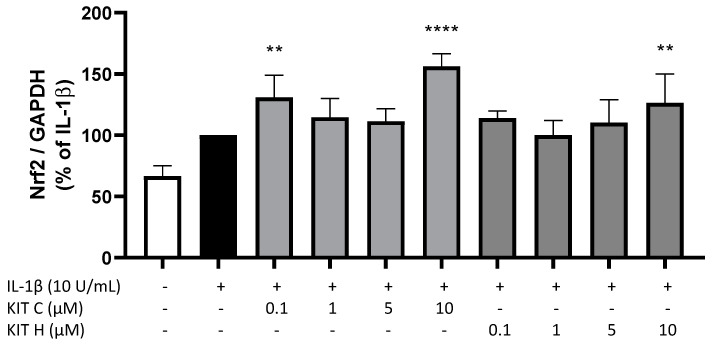
Effects of KIT C (light grey) and KIT H (dark grey) on Nrf-2 expression in IL-1β-stimulated SK-N-SH cells. SK-cells were stimulated as described in material and methods. After 4 h, RNA was isolated and mRNA levels of Nrf-2 was determined using qPCR. Values are presented as the mean ± SD of at least three independent experiments. Statistical analysis was performed using one-way ANOVA with Dunnett’s post hoc tests with ** *p* < 0.01 and **** *p* < 0.0001 compared to IL-1β.

**Figure 7 ijms-22-11665-f007:**
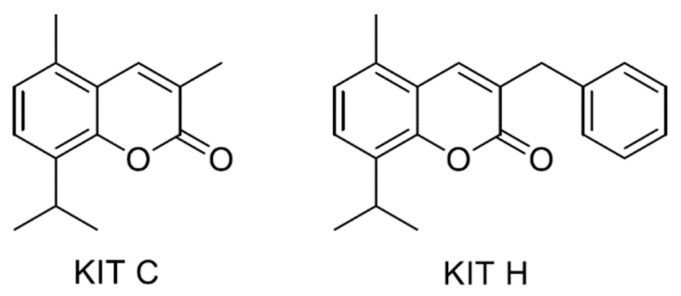
Molecular structure of the synthesized compounds KIT C and KIT H.

**Table 1 ijms-22-11665-t001:** ORAC activity assay data for KIT C and KIT H. Raw values and Trolox™ Equivalents in µM.

Compound	µM TE ^1^	Net AUC ^2^	AUC
Blank	0.0	0.0	1.3684
1 µM KIT C	5.2	0.0743	1.4427
10 µM KIT C	6.5	0.1176	1.4860
1 µM KIT H	3.4	0.0201	1.3885
10 µM KIT H	4.7	0.0607	1.4291

^1^ µM TE = µMole Trolox Equivalents/L; ^2^ Netto Area under the curve blank subtracted (AUC).

## Data Availability

The data presented in this manuscript are available from the corresponding author upon request.

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
