# Peer review of "Targeting Oxidative Stress: Novel Coumarin-Based Inverse Agonists of GPR55"

_ijms, 2021, doi:10.3390/ijms222111665_

Round 1

Reviewer 1 Report

Matthias Apweiler et al. demonstrated the antioxidant effects of two novel synthesized coumarin-based compounds, KIT C and KIT H, in primary mouse microglial and human neuronal SK-N-SK cells.

INTRODUCTION: In general, the Introduction is well described for the part of neurodegenerative diseases and the role of oxidative stress. But, I would like to see consequently, a short introduction for the in-use methods to evaluate oxidative stress in order to better explain the type of radicals measured and the novelty of detecting antioxidants based to the antioxidant source existing in the bibliography.

RESULTS

2.1 Cytotoxic effects of KIT C and KIT H

I would suggest to add statistical significant in the text where necessary.....and the p-value to understand from the text and don't check its time the figure.

2.2 Please explain the importance of this experiment to the hypothesis and also how to measure  8-Iso-PGF2α and not ROS?

2.3 I don't trust 3. ORAC activity assay , I prefer HPLC directly for this kind of measurements. 

2.6. Nrf-2 expression in SK-N-SH cells 

I suggest repeating the statistical analysis for figure 6, doesn't look so correct, if you revise I would like to see the raw data.

Author Response

Matthias Apweiler et al. demonstrated the antioxidant effects of two novel synthesized coumarin-based compounds, KIT C and KIT H, in primary mouse microglial and human neuronal SK-N-SK cells.

Dear reviewer 1,

many thanks for taking the time and reviewing the manuscript “Targeting oxidative stress: Novel coumarin-based inverse agonists of GPR55”.

INTRODUCTION: In general, the Introduction is well described for the part of neurodegenerative diseases and the role of oxidative stress. But, I would like to see consequently, a short introduction for the in-use methods to evaluate oxidative stress in order to better explain the type of radicals measured and the novelty of detecting antioxidants based to the antioxidant source existing in the bibliography. 

Answer: In our manuscript, we focused on inverse agonism as the topic of the Special Issue, existing studies, and clinical implications in the treatment of neurological and psychiatric diseases in the introduction. We described the used methods in the method section and compared and evaluated with other methods in the discussion. Therefore, we did not include the used methods and a detailed introduction for ROS and oxidative stress in the introduction. However, we added some more information for the used methods in the revised version (lines 86-96). Since the types of radicals are not focus of our study, which was the GPR55-dependency of the anti-oxidative compounds used, we kept the revised introduction of oxidative stress short.

2.1 Cytotoxic effects of KIT C and KIT H

I would suggest to add statistical significant in the text where necessary.....and the p-value to understand from the text and don't check its time the figure.

Answer: We now added “significantly” as well as the p-values where applicable.

2.2 Please explain the importance of this experiment to the hypothesis and also how to measure  8-Iso-PGF2α and not ROS?

Answer: For measuring 8-Iso-PGF2a, we used a commercially available enzyme immunoassay (EIA), as described in the method section. However, we added this information in the revised version of the manuscript (line 130). The contribution of the 8-Iso-PGF2a results to our hypothesis is outlined in the discussion.

2.3 I don't trust 3. ORAC activity assay , I prefer HPLC directly for this kind of measurements. 

Answer: For the measurement of antioxidative capacity, we decided to use the ORAC activity assay, since it is well established and often used in research and it was available for us. Since ORAC activity assays and HLPC antioxidative analyses are commonly used in research, both methods can be used based on the availability. Since we do not have access to HPLC, we chose the ORAC assay.  Additonally, we used cell-based methods (such as 8-Iso-PGF2a and ROS-MTT assay) to evaluate the anti-oxidative effects of the coumarin derivates. Therefore, the ORAC activity assay results are not solely used to discuss our hypothesis but are supported by further data.

2.6. Nrf-2 expression in SK-N-SH cells 

I suggest repeating the statistical analysis for figure 6, doesn't look so correct, if you revise I would like to see the raw data.

Answer: We repeated the statistical analysis for Nrf2 qPCR and confirmed the indicated statistical results using GraphPad Prism 8 (One-way ANOVA with following Dunett’s Post-hoc test). However, we used SD in the graphs instead of SEM, we corrected this throughout the revised manuscript. We will submit the raw data (as Word file, please see the attachment) with the revised manuscript as well.

Reviewer 2 Report

current study investigates the antioxidative effects of two novel synthesized cou-92 marin-based compounds, the study is well designed and the results are interesting showing promising use of these compounds in neurological diseases  

Author Response

current study investigates the antioxidative effects of two novel synthesized cou-92 marin-based compounds, the study is well designed and the results are interesting showing promising use of these compounds in neurological diseases  

Dear reviewer 2,

many thanks for taking the time and reviewing the manuscript “Targeting oxidative stress: Novel coumarin-based inverse agonists of GPR55”. We are very happy, that you evaluated our study so positive and that we were able to show the promising potential of the coumarin derivates to you.

Reviewer 3 Report

The paper titled: “Targeting oxidative stress: Novel coumarin-based inverse agonists of GPR55” is well prepared and very interesting for readers. Introduction is written in an understandable and logical way, it well introduces the reader to the subject of research. The results are presented in the Table and Figures what increases the quality of manuscript and make its easy to follow. Additionally, the results have a potential a therapeutic importance in neurological and psychiatric diseases. 
A minor error: 
The aim of the study (lines 86-93) need to be rewritten to be more clarified. 

Author Response

The paper titled: “Targeting oxidative stress: Novel coumarin-based inverse agonists of GPR55” is well prepared and very interesting for readers. Introduction is written in an understandable and logical way, it well introduces the reader to the subject of research. The results are presented in the Table and Figures what increases the quality of manuscript and make its easy to follow. Additionally, the results have a potential a therapeutic importance in neurological and psychiatric diseases.

Dear reviewer 3,

thank you very much for taking the time and reviewing the manuscript “Targeting oxidative stress: Novel coumarin-based inverse agonists of GPR55”.

A minor error:  The aim of the study (lines 86-93) need to be rewritten to be more clarified. 

Answer: Thanks for this good comment, we rewrote the study’s aims to elucidate the idea of the study more comprehensive (lines 97-110).

Round 2

Reviewer 1 Report

Dear Editor and authors,

Unfortunately the authors didn't follow all of my suggestions but some of them. But,  the manuscript is improved and if the authors check better the english language it could be published.